# Overview of Sigma-1R Subcellular Specific Biological Functions and Role in Neuroprotection

**DOI:** 10.3390/ijms24031971

**Published:** 2023-01-19

**Authors:** Véronik Lachance, Sara-Maude Bélanger, Célia Hay, Victoria Le Corvec, Vina Banouvong, Mathieu Lapalme, Khadija Tarmoun, Guillaume Beaucaire, Marc P. Lussier, Saïd Kourrich

**Affiliations:** 1Département des Sciences Biologiques, Université du Québec à Montréal, 141 Avenue du Président-Kennedy, Montréal, QC H2X 3X8, Canada; 2Centre d’Excellence en Recherche sur les Maladies Orphelines-Fondation Courtois, Pavillon des Sciences biologiques, 141 Avenue du Président-Kennedy, Montréal, QC H2X 3Y7, Canada; 3Département de Chimie, Université du Québec à Montréal, 2101, Rue Jeanne-Mance, Montréal, QC H2X 2J6, Canada; 4Center for Studies in Behavioral Neurobiology, Concordia University, Montreal, QC H4B 1R6, Canada

**Keywords:** Sigma-1R, chaperone, calcium, lipid, homeostasis, endoplasmic reticulum, ER stress

## Abstract

For the past several years, fundamental research on Sigma-1R (S1R) protein has unveiled its necessity for maintaining proper cellular homeostasis through modulation of calcium and lipid exchange between the endoplasmic reticulum (ER) and mitochondria, ER-stress response, and many other mechanisms. Most of these processes, such as ER-stress response and autophagy, have been associated with neuroprotective roles. In fact, improving these mechanisms using S1R agonists was beneficial in several brain disorders including neurodegenerative diseases. In this review, we will examine S1R subcellular localization and describe S1R-associated biological activity within these specific compartments, i.e., the Mitochondrion-Associated ER Membrane (MAM), ER–Lipid Droplet (ER–LD) interface, ER–Plasma Membreane (ER–PM) interface, and the Nuclear Envelope (NE). We also discussed how the dysregulation of these pathways contributes to neurodegenerative diseases, while highlighting the cellular mechanisms and key binding partners engaged in these processes.

## 1. Introduction

Neurological disorders currently represent the first leading cause of disability and the second leading cause of death worldwide. Amongst these conditions, neurodegenerative disorders such as Alzheimer’s disease (AD), dementia, Parkinson’s disease (PD), and motor neuron diseases (MNDs) act as major contributors to this global burden [1]. With the continuing aging and growing of populations, this burden is expected to persist, and overwhelm our already overstretched healthcare services even more. Therefore, an urge for the development of new treatments or strategies to prevent, cure, and improve the quality of life of affected patients is of utmost priority. One interesting target that has recently re-emerged and is showing promising outcomes regarding the treatment of many neurological and neurodegenerative disorders, is the Sigma receptors class, including Sigma-1R (S1R) and Sigma-2R/TMEM97 (S2R). Although being also discovered in 1976, S2R identity was very recently defined as TMEM97 [2,3,4], and will not be reviewed herein.

During the past 40 years, many studies have described numerous functions of S1R that help to maintain proper cell homeostasis. Unlike any other protein, S1R is exclusive to mammalian cells and has poor structural homology with other mammalian proteins (although, it shares approximately 30% identity with fungal sterol isomerase, ERG2) [5,6]. S1R was first described as an opioid-receptor [7,8,9], but later referred to as a calcium-sensitive ligand-operated endoplasmic reticulum (ER) chaperone [10]. Nowadays, S1R is recognized as a unique protein unrelated to any traditional classes of opioid receptors given its ability to bind a plethora of ligands, and thus, is officially classified as a non-opioid intracellular receptor [11,12,13]. However, it is important to acknowledge that S1R does not entirely fit the conventional ‘receptor’ definition, given the absence of direct activation of downstream signaling pathways upon ligand-binding. This receptor is ubiquitously expressed but shows higher protein expression in the liver, placenta, and brain (especially in cerebellum, hippocampus, basal ganglia, and cerebral cortex) [14]. Importantly, brain transcriptomic data suggest that S1R expression is higher in glial cells than neurons [15]. Modulation of S1R activity is beneficial in various models of neurological and psychiatric disorders, including but not limited to AD, PD, MND, Huntington’s disease (HD), anxiety, and depression. Moreover, genetic variants of the SIGMAR1 gene have been discovered and linked to neurodegenerative and neuromuscular disorders (reviewed in [16,17,18]), therefore, highlighting the importance of this receptor in preserving healthy brain functions. As of today, many human clinical trials targeting S1R activity in neuropsychiatric disorders are under investigation (reviewed in [19]). Herein, we will review our understanding of S1R’s biological functions and its neuroprotective role, while briefly discussing its dysregulation in neurodegenerative disorders, and current promising drug candidates.

## 2. Molecular Mechanisms Regulated by S1R

ER represents the largest and the most multifunctional organelle in cells. Thus, it is not surprising that S1R has been associated with so many mechanisms (e.g., ER-stress response, Ca^2+^ homeostasis, lipid metabolism, autophagy, etc.). To better understand S1R biological functions, one must understand its ‘living’ environment, hence the ER structures, its organization, and functions. As a continuous interconnected membrane network, the ER shows two major domains including the nuclear envelope (NE) and the peripheral ER. The NE is composed of two flat ER membrane bilayers that once stacked, will create the inner and outer nuclear membrane (INM and ONM, respectively) [20,21]. Morphological studies of the peripheral ER were first visualized by electron microscopy and classified into rough ER (RER) and smooth ER (SER). On the one hand, the RER was characterized by the presence of ribosomes bound to the ER membrane and is now referred to as ER sheets. On the other hand, the SER was defined by the absence of ribosomes on its membrane and is now referred to as ER tubules [22,23,24,25,26]. Both ER sheets and tubules are derived from the NE, more precisely the ONM [22,27]. Despite being interconnected, the NE, the ER sheets, and tubules are all associated with very distinct functions and are spatially separated, explaining the diverse biological functions of S1R. The following sections will discuss the role of S1R in ER sheets, tubules, and NE, while shedding light on the chaperone’s localization at different membrane contact sites and its impact on inter-organelle communication (Figure 1).

### 2.1. S1R and ER-Tubule-Related Functions

#### 2.1.1. Mitochondrion-Associated ER Membrane (MAM)

##### Calcium Homeostasis and Lipid Metabolism

Even before being cloned and sequenced in 1996 [5], S1R was suspected of being an ER-resident protein since its ligand-binding sites were mostly detected in microsomal fraction (an ER-enriched fraction) during subcellular fractionation experiments [28,29]. Given that the ER critically regulates Ca^2+^ level, a pioneer study linked S1R activity to calcium homeostasis by showing that haloperidol, an antagonist of S1R, enhanced the concentration of free cytosolic Ca^2+^ and triggered cell apoptosis in adenocarcinoma cell lines [30]. Consequently, Brent et al.’s study [30] has prompted others to examine the relation between S1R and the inositol-triphosphate-receptor (IP3R), a key player in the modulation of intracellular Ca^2+^ level, and a well-known ER-resident protein. A brief report first showed that inhibition of IP3R with heparin decreased the agonist (+)-SKF-10047 binding to S1R, suggesting that both receptors were in close vicinity [31]. Altogether, these studies were the first to support the role of S1R in the regulation of intracellular Ca^2+^ through a potential ER-linked pathway that might involve IP3R activity. Additional studies have next demonstrated S1R agonist-induced modulation of Ca^2+^ signaling via two modes of action: one involving intracellular components, most likely IP3R, and the other involving G-protein-dependent action at the plasma membrane [32]. S1R association with the type 3 IP3R (IP3R3) was finally demonstrated in the early 2000s using co-immunoprecipitation and co-localization analyses in NG108 cells (neuronal-like cells) [33]. In this study, the authors demonstrated S1R interaction with IP3R3, Ankyrin 220, and Ankyrin 135 (ANK220 and ANK135, respectively). Interestingly, agonist-activation of S1R promoted ANK220, but not ANK135, dissociation from the IP3R3, which potentiated Ca^2+^ efflux from the ER upon bradykinin-induced Ca^2+^ release (Figure 2) [33].

Currently, it is well accepted that S1R prevalent localization is at the MAM within the ER-tubule network [10,34]. In fact, it has been shown that, together with the ER chaperone BiP (Binding Immunoglobulin Protein, also known as Grp78 or Hspa5), S1R forms a Ca^2+^ sensitive complex at the MAM. Upon ER stress, such as Ca^2+^ depletion or agonist stimulation, S1R dissociates from BiP to associate with, and to stabilize, the IP3R3 at the MAM interface. By doing so, S1R nurtures Ca^2+^ transfer from ER to mitochondria through an IP3R3-dependent pathway [10]. Importantly, the agonist-induced activity of S1R does not upregulate the Ca^2+^ level itself. However, once bound to an agonist, S1R can potentiate calcium signaling following cell stress events. Further characterization of S1R localization at the MAM has indicated that S1R is predominantly confined to lipid raft-like detergent-resistant microdomains (DRMs). Within MAM-derived DRMs, S1R was shown to associate preferentially with sphingolipids, such as ceramides (lactosylceramide > galactosylceramide > glucosylceramide > sulfatide) and sterols (lathosterol > progesterone > testosterone > cholesterol) in Chinese hamster ovary (CHO) cells [35]. Interestingly, ceramide and cholesterol depletion, with Fumonisin B1 and methyl-β-cyclodextrin, impaired S1R translocation at MAM-derived DRMs, suggesting that anchoring of S1R at this peculiar position is lipid-dependent [35]. Moreover, S1R depletion was shown to reduce mitochondrial pregnenolone synthesis in MA-10 cells (mouse Leydig cell line) [36]. In their study, Marriott et al. proposed that S1R and VDAC2 (voltage-dependent anion channel 2) interaction is mediated through StAR (steroidogenic acute regulatory protein), and that S1R-StAR association occurs before StAR import into mitochondria. The formation of the VDAC2–S1R–StAR protein complex was then proposed to support cholesterol trafficking at the MAM, and to promote mitochondrial pregnenolone synthesis [36] (Figure 2). However, further investigation regarding S1R role in steroidogenesis is required. Knowing that pregnenolone [37] itself or its derivatives, i.e., DHEA [37] and progesterone [12], are S1R agonists and antagonists, respectively, it would be interesting to verify if S1R activity in this pathway is regulated through a retro-feedback activation or inactivation loop. It would also be of interest to determine if S1R, itself, is able to modulate the activity of converting enzymes involved in this pathway. S1R regulation of these enzymes’ activity is more than likely possible given the recent discovery of S1R as an integral mitochondrial membrane resident protein in adult mouse cardiomyocytes [38].

##### Autophagy

Another mechanism initiated at the MAM [39], critical for maintaining cell homeostasis and cell survival, is autophagy. This lysosomal catabolic process is essential for recycling or removing damaged organelles, for breaking down toxic protein aggregates, and for clearing pathogens. It is a multi-step process that requires the nucleation of a phagophore, that originates from the omegasome (a phosphatidylinositol-3-phosphate enriched ER membrane) and the MAM. During the elongation step, the phagophore matures into a double membrane autophagosome (AP), while engulfing cargos that will later be degraded upon AP fusion with lysosomes (autolysosomes) (Figure 3) [40]. Dysregulation of autophagic activity has been associated with many neurodegenerative diseases [41]. Interest in the S1R role in autophagy regulation increased when it was first shown to co-localize with protein aggregates and autophagy markers (such as p62/SQSTM1 and LC3) in the brain and cells of patients affected by neurodegenerative disorders [42,43]. In addition, a genetic variant of S1R (p.E102Q) has been linked to juvenile amyotrophic lateral sclerosis (ALS) and was shown to impair APs fusion with the lysosome and thus, to enhance the accumulation of APs within cells [43]. Aggregation of autophagy markers was also associated with two S1R variants (p.E138Q and p.E150K) causing distal hereditary motor neuropathies (dHMNs). Both variants were shown to impair ER-mitochondria tethering and Ca^2+^ signalling in SH-SY5Y [44]. Yang et al. (2019) later showed S1R involvement in mitophagy (selective autophagic degradation of mitochondria). Through interaction with ATG14, STX17, and VAMP8 (key proteins involved in autophagosome–lysosome fusion), S1R was shown to increase APs fusion with the lysosome in multiple models (mouse retina, NSC34, SH-SY5Y, and HEK293 cells) [45].

Interestingly, Wang et al. also revealed that S1R expression or S1R agonist (pridopidine) indirectly induced autophagy by facilitating TFEB nuclear translocation, hence supporting TFEB-mediated autophagy-related gene (ATG) transcription and lysosome biogenesis [46]. Agonist-induced activity of S1R with ANAVEX2-73 was also shown to increase autophagy flux in HeLa cells and *C. elegans* [47]. However, deeper characterization of the mechanisms driven by S1R during autophagosome fusion and autophagy induction is required. In fact, it would be interesting to understand which autophagy signal is promoting S1R action within the pathway, and if these upstream signals can guide S1R involvement toward bulk autophagy or selective autophagy. Given that ANAVEX and pridopidine induce autophagy through different pathways, it would be of great interest to better characterize these compound effects on upstream autophagy regulators.

Altogether, the impact of S1R in calcium signaling and autophagy defines the necessity of S1R functions in supporting mitochondria metabolism and maintaining MAM integrity. It demonstrates that S1R localization at the MAM facilitates its binding to BiP (and other partners) and allows it to quickly sense changes in Ca^2+^ homeostasis (or other ER-stressors), and to effectively promote, for instance, cell survival signaling pathways or steroidogenesis by supporting ER-mitochondria exchange (e.g., Ca^2+^ and cholesterol) and autophagy.

#### 2.1.2. ER-Lipid Droplet (ER-LD) and Lipid Metabolism

S1R was also shown to be confined to the raft-like enriched-microdomain of the ER-LD subregion in NG108 cells, where it can modulate lipid compartmentalization and export [48]. Noteworthy is a study that demonstrated that N-terminally tagged S1R (EYFP-S1R), even if remaining at the ER, failed to be targeted to this contact site. Most importantly, only the C-terminally tagged S1R (S1R-EYFP) showed a similar subcellular localization pattern when compared to the endogenously expressed protein. A lack of S1R at this ER-LD contact site was shown to promote an uncontrolled distribution of neutral lipids and cholesterol within the cells. Finally, it was determined that S1R targeting to this compartment was sensitive to brefeldin A treatment (an ER-to-Golgi trafficking inhibitor) [48]. Together, these observations suggest that S1R targeting to the ER-LD microdomain is dependent on appropriate ER-to-Golgi trafficking, while the integrity of the NH_2_-terminal region is detrimental to S1R proper subcellular compartmentalization (Figure 4). In agreement with these findings, a study in PC12 cells has also revealed that S1R expression caused the remodeling of lipid rafts by modifying the ratio of key lipid components, such as gangliosides and cholesterol [49]. S1R effect on lipid-raft composition was later proposed to induce EGFR enrichment and signaling at the non-raft region, which subsequently improved neuritogenesis [49]. Akin to this observation, another study has determined that S1R forms galactosylceramide (GalCer)-enriched lipid rafts at ER-LD-like structures in the myelin sheet of rat mature oligodendrocytes (OLs) [50]. In an OLs progenitor cell line (CG-4 cells), the enriched S1R and GalCer lipid rafts were found to promote the differentiation of OLs [50]. However, how S1R can convert or enrich certain classes of lipids at specific membrane contact sites remains unclear.

Further investigations have demonstrated S1R direct binding to cholesterol through two different cholesterol-binding domains (CBDs), both defined within S1R drug-binding region, and localized at the C-terminal [51]. On the one hand, S1R association with cholesterol in adenocarcinoma cells (MDA-MB-231) was shown to modulate the protein content of rafts (such as integrins), and thereby affected cell adhesion via the β1 integrin–mediated pathway [51]. In fact, agonist-induced activity of S1R with (+) SKF-10047 (which most likely antagonizes the cholesterol binding to S1R) or its depletion were both shown to reduce the amount of cholesterol incorporated in lipid rafts. This phenomenon was later found to diminish β1-integrin recruitment to lipid rafts and to impair cell adhesion [51]. In this study, the authors finally hypothesized that S1R ability to remodel the lipid and protein composition of the lipid rafts might help cancer cells to stimulate oncogenic signaling pathways, and suggested that the use of S1R drugs could destabilize the rafts and help to better sensitize these cells to apoptotic agents.

In agreement with this report, S1R binding to cholesterol was confirmed and further proved to promote S1R clustering, and was shown to increase the membrane thickness using in vitro giant unilamellar vesicles (GUVs) reconstitution assays. Interestingly, S1R activation with (+) SKF-10047 reduces the amount and size of the clusters [52]. While validating Y201 and Y206 residues’ interaction with cholesterol as previously predicted [51], the authors also identified a new CBD within the transmembrane (TM) region of the chaperone. These S1R-cholesterol-enriched microdomains were found to improve the recruitment of the inositol-requiring enzyme 1a (IRE1α) and its clustering in GUVs. Deletion of S1R expression impaired IRE1α signaling in HEK293 cells, again supporting the idea that S1R’s ability to remodel (and/or stabilize?) the lipid raft may provide the cells with a new platform able to initiate specific signaling cascades, or to facilitate the contact between ER membrane and other organelles [52,53]. Interestingly, the authors also reported that point mutation or deletion of the di-arginine motif (RR) within the NH_2_-terminal reversed S1R orientation in HEK293 cells, meaning that the C-terminal moiety of the chaperone was no longer facing the ER lumen, but was rather exposed to the ER cytosolic face [52]. This observation indicates once again the importance of S1R N-terminal integrity, not only for its proper intracellular compartmentalization, but also for its orientation. It also suggests that N-terminal cleavage or post-translational modifications, such as phosphorylation, citrullination or methylation, could deeply modify S1R biological functions. Intriguingly, no report was found regarding post-translational regulation of S1R biological activity, and thus, effort should be put toward that question. Discrepancies in S1R N- and C-terminal orientation were also observed when comparing lipidic cubic phase purified crystal structure to electron microscopy photomicrographs from in cellula and in vivo transient expression of S1R-GFP-APEX2 [6,54]. Given the ambiguity regarding S1R orientation in cells, one should keep in mind that the chaperone positioning might also be affected depending on the organisms used and the lipid composition of the membrane [55,56,57]. However, most studies agree that S1R has a unique transmembrane, an intra-luminal carboxy-terminal, and a cytosolic amino-terminal domain.

In summary, S1R seems able to modulate the lipid raft composition by directly binding specific lipids (such as ceramide and cholesterol). The lipid incorporated in the raft appears to vary depending on the cell type, and may potentially be organelle specific. In fact, one could propose that changes in the lipid constituents of the rafts influence S1R integration to it. On the other hand, one could also propose that S1R recruitment to the raft changes its lipid signature, which consequently induces new signaling pathways or recruits new binding partners. However, which event occurs upstream or downstream and how these phenomena are triggered remains ambiguous.

### 2.2. S1R and ER-Sheet Related Functions

#### 2.2.1. ER-Plasma Membrane (ER-PM) and Calcium Entry

Many studies have observed and reported S1R translocation to the ER-PM junctions upon agonist-binding (reviewed in [58,59]). In fact, S1R is known to affect the expression and stabilization of many receptors, such as ion channels, kinases, GPCRs (G Protein-Coupled Receptors), and integrins (review in [59]). This wide range of interacting partners allows S1R to act as a very potent modulator of cellular excitability, adhesion, growth, inflammation, and signaling. Moreover, S1R was also shown to inhibit store-operated Ca^2+^ entry (SOCE) through association with STIM1 (Stromal Interaction Molecule 1), an ER-Ca^2+^ sensor regulating SOCE [60]. Indeed, it was demonstrated that evoked Ca^2+^ release, with ER stressors such as thapsigargin, carbachol, ionomycin, or cyclopiazonic acid (CPA), was reduced in HEK 293 cells stably expressing S1R, when compared to WT (wild type) cells. These results suggested that ER calcium level was lower in S1R stable cells. Interestingly, upon Ca^2+^ restoration, S1R cells showed a reduced ability to replenish their intracellular Ca^2+^ concentration. Likewise, these observations were reproduced using S1R agonist SKF-10047. However, S1R antagonist (BD-1047) and S1R depletion using small interfering RNA (siRNA) showed opposite effects, i.e., it enhanced extracellular calcium reuptake. Of note, S1R depletion highly decreased Orai1 expression, while increasing STIM1 expression level. In this study, the authors confirmed the S1R interaction with STIM1 by using co-immunoprecipitation assay, confocal microscopy, and AFM, and showed that agonist stimulation enhanced S1R association to STIM1 and that S1R-STIM1 complex was translocated to ER-PM junctions upon Ca^2+^ store depletion. Once at the ER-PM, S1R binding to STIM interfered with Orai1-STIM1 association and hence, reduced SOCE activity (Figure 5) [60]. These data unveiled that S1R is not only able to modulate calcium homeostasis between ER and mitochondria, but it can also control the cell’s ER Ca^2+^ concentration and Ca^2+^ reuptake after a prolonged stress period. The siRNA experiments performed also offered a hint regarding S1R effect on Orai1 expression. Although shown to associate with Orai1, S1R’s impact on the Ca^2+^-channel stability and expression at the PM remains to be clarified. Interestingly, another study using breast cancer cell line (MDA-MB-435s) and colorectal cancer cells (HCT-116) also reported that S1R activation (with Igmesine) was reducing extracellular calcium reuptake in a SK3-dependent manner, most likely by inhibiting Orai1 translocation in DRMs. Most importantly, disruption of calcium signaling was later found to impair cancer cell migration [61]. Briefly, S1R can promote the cell survival pathway by supporting ER-mitochondria exchange, and is likely able to protect cells from a Ca^2+^ overload through SOCE regulation at the ER-PM junctions. These findings suggest that S1R can prevent hyperactivation of Ca^2+^-operated enzymes, and most likely regulate cell migration and hyperexcitability (reviewed in [62]).

#### 2.2.2. Chaperone Activity and ER Stress

Molecular chaperones are defined as proteins having the ability to assist other macromolecules’ folding and maturation, and to prevent their misfolding and aggregation per se [63]. ATP-dependent chaperones comprise Hsp60 (chaperonins), Hsp70 (such as BiP), Hsp90, and Hsp100 (Clp proteins) families, whereas ATP-independent chaperones include Hsp40 (J-proteins) and sHsp (small heat shocks proteins) families. As expected from a molecular chaperone, S1R’s expression level is enhanced upon cellular stress such as heat shock treatment, glucose deprivation, and induced ER stress with tunicamycin and thapsigargin [10]. Hayashi et al. (2007) revealed a strong increase in S1R expression after 30 min of stimulation with each stressor mentioned above, and a maximal effect was reached after 2–3 h of exposure. Interestingly, long-term glucose deprivation and tunicamycin treatment turned down S1R expression, while heat shock and thapsigargin stimulation both prolonged S1R expression level over time [10]. This highlights that the nature of the stressors can specifically modulate S1R expression level over time. Moreover, one response seems to involve an energy-sensing pathway and maintenance of the proteostasis network, while the other involves calcium homeostasis and a temperature-sensing pathway. Furthermore, S1R capacity in preventing protein aggregation was shown using ATP-independent light-scattering analysis of citrate synthase aggregation assay [10,46]. These findings suggest that S1R can be categorized as an ATP-independent chaperone. However, considering that its association with BiP was characterized as dormant [10], it invalidates its role as a co-chaperone (like Hsp40 families). Given that very few groups have investigated the concept, this idea needs to be revisited. Worthy of note, all sHsp share the following features: (i) they bind, stabilize and prevent non-native protein aggregation, (ii) maintain protein homeostasis in an ATP-independent manner, (iii) the monomer has a small molecular weight (range from 12–42 kDa), (iv) they are induced by stress conditions, (v) usually form large and “inactive” oligomers, (vi) have a dynamic quaternary structure, and (vii) have a conserved α-crystallin domain [64,65]. Out of these seven features, six of them perfectly align with our knowledge regarding S1R biological activity. Therefore, this allows us to categorize S1R as a non-canonical sHsp given its unique structure and ligand-inducible properties.

Through ER stress sensors, cells can verify the status of their proteome integrity using different ER quality control mechanisms. To counteract ER stress and restore homeostasis, activation of ERAD (ER-associated degradation), UPR (unfolded protein response), and ER-phagy (selective degradation of ER by autophagy) is required [66,67]. ERAD is known to be the main quality-control mechanism accountable for misfolded protein degradation using the cytosolic ubiquitin-proteasome system (UPS) [68]. Components of the ERAD include Hrd1, Sel1L, and p97 (VCP/CDC48). In general, ERAD substrates can be recruited by several chaperones in the ER, such as BiP, and are next delivered to the Sel1L–Hrd1 complex. This complex is the most conserved ERAD complex in mammals. It promotes the retro-translocation of the misfolded substrate to the cytosol. Once in the cytosol, the substrate is ubiquitinated by p97, and is subsequently subjected to proteasomal degradation [68]. In an HD cell model, it was shown that S1R was linked to proteolysis of intranuclear huntingtin (Htt) protein aggregates through an ERAD-associated mechanism and not autophagy [69]. However, a deeper characterization of the ERAD machinery involved in this phenomenon is required. Given the recent identification of the INMAD (inner-nuclear membrane-associated degradation) mechanism, it would be interesting to investigate the role of S1R in this phenomenon [70,71]. ERAD is also important for sterol homeostasis given that it is involved in HMGR (HMG-CoA reductase, the rate-limiting enzyme that produces cholesterol) degradation through Insigs (Insulin-induced genes) activity. At the mechanistic level, Insigs regulate sterol homeostasis using two pathways, both triggered by the level of intracellular cholesterol: (i) Insigs–HMGR complex enhances ERAD-dependent degradation using gp78–p97 machinery, while (ii) Insig-Scap complex blocks Scap-SREBP transport to the Golgi and thus, inhibits the SREBP-dependent activation of gene transcription crucial for cholesterol synthesis [72]. Interestingly, S1R was found to associate with Insig1 in an agonist and sterol-sensitive manner [73]. In this study, the authors have shown that S1R–Insig1 complex formation was enhanced by S1R agonist or Insig1 binding to 25-hydroxycholesterol (25-HC). This complex was next found to enhance HGMR degradation and to reduce Insig1 and Scap interaction. Interestingly, the authors further propose that S1R–Insigs interaction might also drive ceramide galactosyltransferase (CGalT) degradation through a similar pathway. S1R colocalized with CGalT and seemed to reduce galactosylceramide (GalCer) synthesis [73]. This study further supports the action of S1R in lipid metabolism, and unveils its function in Insig-dependent ERAD.

The adaptive response (pro-survival) mediated by the UPR allows cells to establish communication between the ER and the nucleus, and to react appropriately to a buildup of misfolded protein and other stressors. UPR involves different responses that are mediated by three ER stress sensors: IRE1 (inositol-requiring enzyme 1), PERK (protein kinase R (PKR)-like ER kinase), and ATF6 (activating transcription factor 6). Under basal conditions, these factors, like S1R, are all associated in an inactive state with BiP. Upon stress induction, they dissociate from BiP and activate different pathways that will (i) repress protein synthesis (PERK-eIF2 signaling), (ii) increase protein folding (IRE1-XBP1 and ATF6-p50-XBP1 signaling), or (iii) enhance clearance of misfolded proteins (IRE1, PERK and ATF6 signaling pathways) [67,74]. Overall, these sensors activate the transcription and translation of chaperones or antioxidant proteins to restore ER homeostasis. However, if it fails to reach homeostasis, overactivation of these pathways will lead to a maladaptive stress response (pro-apoptotic) that will enhance the expression of CHOP (CCAAT/enhance binding protein (C/EBP) homologous protein) and induce apoptosis (PERK-eIF2-ATF4-CHOP and ATF6-CHOP signaling) [74]. Interestingly, S1R and IRE1 have been shown to localize and associate at the MAM in cells [75]. S1R seems to stabilize unfolded IRE1 monomer at the MAM, and most likely promotes its dimerization and activation when ER stress is induced with thapsigargin. Moreover, S1R can also reduce CHOP nuclear expression after prolonged exposure to tunicamycin, while also promoting IRE1 phosphorylation and increasing XPB1 nuclear expression [76]. Loss of S1R expression in zebrafish also supports its critical role in UPR activation and mitochondria activity [77]. Although being shown to interact with IRE1, it would be interesting to determine if S1R is involved in the nucleocytoplasmic shuttling of CHOP and XPB1.

In summary, numerous studies have proven that S1R functions are more often observed following ER stress. Whether through calcium depletion (thapsigargin) or accumulation of misfolded protein (tunicamycin or aggregation-prone proteins expression), these findings demonstrate that S1R activity promotes pro-survival and cytoprotective mechanisms.

#### 2.2.3. Intracellular Trafficking

Intracellular protein trafficking is of critical importance for the maintenance of cellular homeostasis. Disruption of protein segregation within cells can lead, for example, to the accumulation of misfolded or mislocalized proteins, a phenomenon that is closely associated with the etiology of various neurodegenerative diseases [78]. As mentioned, agonist-bound S1R has been shown to redistribute from the MAM to the ER-PM junctions (reviewed in [58]) or to NE [48,79,80]. However, the mechanism promoting this change in the location remains poorly understood. In fact, very few studies have investigated this phenomenon and were able to give us a clue regarding the potential mechanism related to S1R translocation to a different subcellular compartment. A pioneer study from Tsai et al. (2009) revealed that S1R knockdown indirectly reduces Rac1 GTPase activation by disrupting Tiam1 (a Rac1 GEF (guanine nucleotide exchange factors)) localization in lipid rafts [81,82]. Next, Rac1 and S1R interaction was confirmed and shown to be strengthened by S1R agonist, pentazocine, and GTP-bound Rac1 using isolated cortical non-synaptosomal bovine mitochondria extracts [83]. Together, these studies highlighted for the first time the relation between S1R and small GTPases. Importantly, Rac1 is part of the Rho GTPase family, whose functions within a cell is to control the organization of the actin cytoskeleton and microtubules dynamics [82,84]. Hence, this provides us with the first piece of information regarding the S1R translocation puzzle, i.e., S1R interaction with the active form of Rac1 may enable it to move along the actin cytoskeleton or to reorganize it. Recently, Nakamura et al. showed that the S1R–Arf6 complex controls extracellular vesicle (EVs) secretion using a cocaine-dependent signaling pathway [85]. The authors found that the S1R–Arf6 complex was enriched at the MAM, and that S1R preferentially binds to the GDP-bound Arf6 (inactive form), which makes S1R a possible regulator of Arf6 activity. Upon cocaine exposure (S1R agonist), the S1R–Arf6 complex is disrupted, which allows Arf6 to promote the secretion of EVs. This study suggests that S1R activation also enhances Arf6 activity and indirectly promotes EVs trafficking through an actin-dependent mechanism. However, further analysis is required to claim S1R as an Arf6-GEF (guanine nucleotide exchange factor).

Arf6 localizes at the plasma membrane, cytosol, and other endosomal membranes (e.g., endosomes, recycling endosomes) and activates multiple lipid modifying enzymes [86]. It can also promote Rac1 activity by inducing Rac1-GEF functions [87]. In this context, it will be of great interest to further characterize S1R–Arf6–Rac1 interaction, signaling, and trafficking pathways, which could help us to better understand S1R sorting to so many subcellular compartments.

### 2.3. S1R and NE-Associated Functions

#### 2.3.1. Gene Transcription and mRNA Translation

Very few studies have investigated the functions of S1R at the NE. An original study on this subject showed that upon agonist exposure, i.e., (+)-pentazocine and cocaine, S1R was able to translocate from ER-LD to NE in NG108 cells [79]. S1R localization at the NE was further supported by an electron microscopy study showing S1R mainly located at the NE in mouse retinal neurons (including photoreceptor, bipolar, and ganglion cells) [88]. Later, it was established that upon cocaine stimulation, S1R translocation to NE enabled it to recruit chromatin-remodeling components, including Emerin, Lamin A/C, BAF (barrier-to-autointegration factor), and HDAC (histone deacetylase). Recruitment of these components was found to repress MAOB (monoamine oxidase B) gene expression through association with the transcription factor Sp3 (specific protein 3), in NG108 and Neuro2a cells (Figure 6) [80]. Interestingly, S1R modulation of gene transcription and mRNA translation was also described in neurons of dorsal root ganglia (DRG) in a spare nerve injury model of neuropathic pain [89]. In this study, the authors proposed that S1R translocation from ER to NE is supported by Sec61β, a transport protein mainly residing at the ER (Figure 6). Once at the nucleus, S1R was shown to interact with c-Fos and to bind 4E-BP1’s promoter, thereby enhancing its expression and indirectly reducing Cav2.2 mRNA translation through inhibition of eIF4E [89].

#### 2.3.2. Nucleocytoplasmic Transport and Clearance of Nuclear Inclusions (NIs)

A previous study has shown S1R co-localization with Ran BP2 (also known as nucleoporin 358, Nup358), which suggests that S1R might be found at the nuclear pore (NP) [80]. Most recently, Lee et al. (2020) demonstrated that S1R was able to interact with many nucleoporins (Nup50, Nup62, and Nup358) and Ran GAP (Ran GTPase activating protein) in HeLa cells and NSC-34 motor neuron-like cells [90]. S1R association with Nups was shown to enhance their stability, and to improve Ran GTPase nucleocytoplasmic distribution in an ALS/FTD cell line model, i.e., HeLa or NSC-34 cells overexpressing (G4C2)31-RNA repeats. Consequently, S1R was suggested to facilitate nucleocytoplasmic transport through stabilization of the nuclear pore complex (NPC), and by doing so, to ease Ran GTPases trafficking between the nucleus and the cytoplasm [90]. Consistent with this study, Wang et al. (2022) also demonstrated the association of S1R with POM121 (another nucleoporin) and KPNB1/Importinβ1 at the nuclear pore in NSC-34 cells [46]. Overexpression of S1R in cells overexpressing (G4C2)31-RNA repeats or treated with S1R-agonist (i.e., pridopidine) was found to stabilize POM121, but did not change KPNB1/Importinβ1 stability. Stabilization of the POM121–KPNB1/Importinβ1 complex through S1R expression or pridopidine treatment was further shown to restore KPNB1/Importinβ1 and TFEB translocation to the nucleus (Figure 5) [46]. Interestingly, S1R-positive NIs were observed in neurons of patients affected by polyglutamine diseases (including HD, spinocerebellar ataxia (SCA), and others), suggesting that S1R might, itself, shuttle between the nucleus and cytoplasm and could participate in ER-related degradation of neuronal Nis [42]. This assumption was later confirmed while showing an increased formation of NIs in the S1R-depleted HD cell model, or in cells treated with the proteasomal inhibitor, i.e., epoxomicin. On the other hand, HD cells overexpressing S1R showed a reduced number of NI, further supporting the role of S1R in ER-associated clearance of NIs [69].

Taken together, these studies demonstrated S1R localization at the NE and shed some light on the role of S1R within this organelle. Indeed, S1R scaffolding and chaperoning abilities not only promote and stabilize the NPC assembly, but also facilitate the nucleocytoplasmic transport, and recruitment of chromatin-remodeling components. Consequently, expression of S1R or S1R-agonist exposure (cocaine, pentazocine, and pridopidine) were proven efficient to maintain nuclear homeostasis and to modulate gene expression and protein synthesis, by directly or indirectly regulating the functions of transcription factors (such as c-Fos, TFEB) and translation initiation factor (such as 4E-BP1).

## 3. S1R as a Therapeutic Target & Conclusions

Pharmaceutical interest towards S1R has re-emerged given the recent advances in S1R crystal structure [6], our improved understanding of its ligand recognition/association [91,92], and biological functions. As discussed above, S1R activity in calcium homeostasis, lipid metabolism, protein folding, autophagy, ER-stress response, intracellular trafficking, gene transcription and translation, makes it a powerful target when trying to rescue or enhance any of these mechanisms that might be dysregulated in a disease. In fact, S1R is now, more than ever, considered a very potent modulator of neuroprotection and a lot of efforts are put together to take advantage of these functions to prevent, stabilize, or even modify neurodegeneration. Thus, we must mention that S1R is a therapeutic target for diseases, including AD [93], PD [94], ALS [95,96], and cancer [97]. In fact, there are ongoing phase 2/3 clinical trials for the treatment of Alzheimer’s disease (NCT03790709, NCT04314934) and phase 2 trials for Parkinson’s disease dementia (NCT04575259). These trials are testing ANAVEX2-73 (also known as Blacarmesine), another compound designed as an S1R agonist (and a muscarinic modulator of M1 receptor and a M2/M3 receptors antagonist) [98]. This drug is involved in the modulation of glutamate release, suppression of neuroinflammation, and restoration of cellular functions essential for maintaining neuronal homeostasis processes such as protein folding, calcium regulation, oxidative stress, ER stress, and autophagy [18,19,47,99]. ANAVEX2-73 has shown not only neuroprotective effects but also anticonvulsant, antiamnesic, and antidepressant properties in various animal models [98,99]. Furthermore, pridopidine (also known as ACR16 or Huntexil^®^) is currently being investigated in a phase 2 clinical trial for the treatment of ALS (NCT04615923). Initially developed for the treatment of Huntington’s disease (HD) associated motor symptoms (reviewed in [18,100]), pridopidine has been shown to protect cells from apoptosis and to improve motor function in an HD mouse model (R6/2). It was shown to: (i) rescue mitochondrial functions and to preserve MAM integrity in human neural stem cells and an HD mouse model (YAC128) [101], (ii) reduce ER stress by modulating all branches of the UPR response in a mHtt (mutant huntingtin) cell line [102], and (iii) protect neurons from mHTT toxicity to decrease cell death [103]. Pridopidine was also shown to decrease neuron death, to conserve neuro-muscular junctions (NMJ), and to restore and enhance axonal transport in primary myocytes and motoneurons cell culture derived from wild-type and SOD1-G93A mice (ALS mouse model) [96]. As of now, targeting S1R activity in neurodegenerative disorders shows great potential. However, combining S1R agonists with other treatments needs to be considered for further analysis of beneficial outcomes. Finally, given that neurosteroids can bind to S1R and modulate its activity, more studies are required to fully understand the effect of aging and biological sex on S1R activity. These studies could help to better estimate S1R therapeutic potency when evaluating its functions in age- and sex-related neurodegenerative disorders.

## Figures and Tables

**Figure 1 ijms-24-01971-f001:**
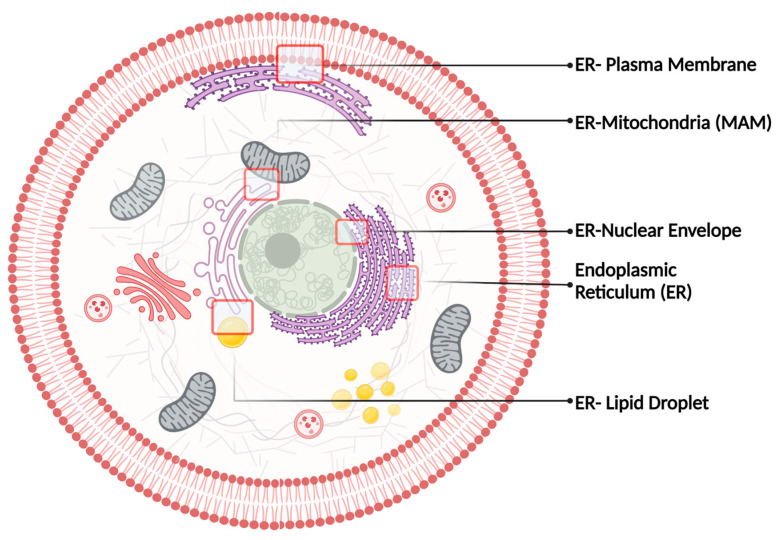
Intracellular membrane contact sites known to harbor S1R.

**Figure 2 ijms-24-01971-f002:**
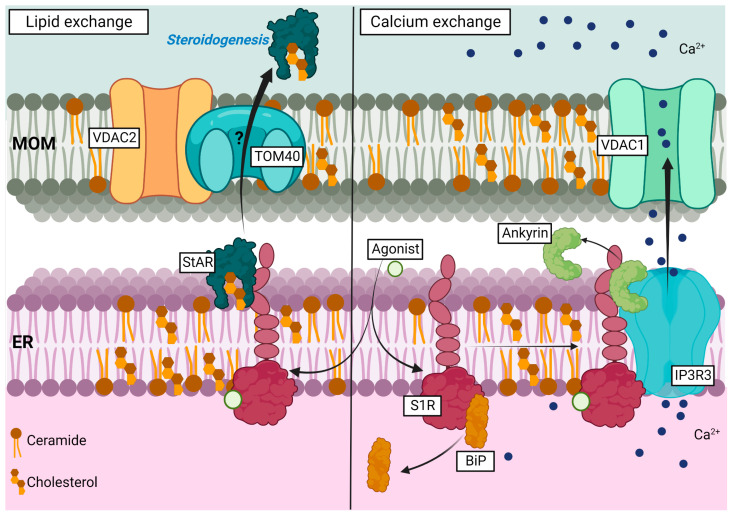
Representation of S1R role in lipid and calcium exchanges observed in lipid raft at the MAM. Left side; agonist-induced activity of S1R promotes its interaction with StAR and supports cholesterol exchange between ER and mitochondria through interaction with VDAC2 and potentially TOM40, which enhances steroidogenesis. Right side: Upon agonist-binding, S1R dissociates from BiP and associates with IP3R3. S1R binding to IP3R3 promotes Ankyrin protein dissociation’s from IP3R3 and supports calcium exchange from the ER to mitochondria. MOM (mitochondria outer membrane).

**Figure 3 ijms-24-01971-f003:**
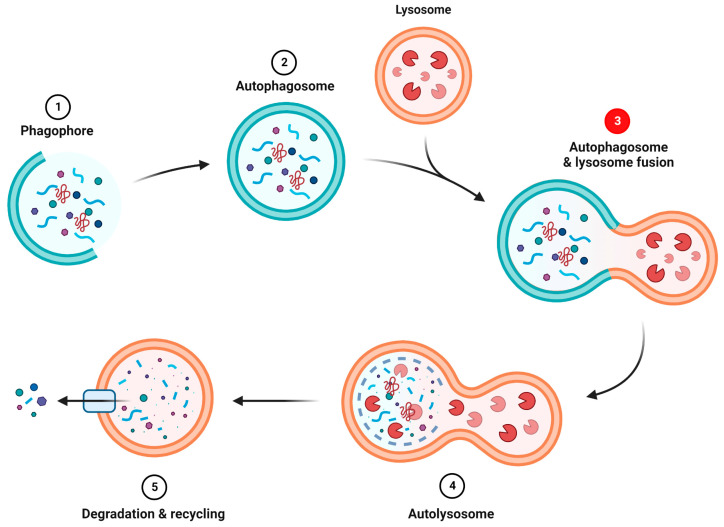
Summary of autophagy process. (1) Nucleation of the phagophore. (2) Elongation and maturation of the phagophore into an autophagosome. (3) Autophagosome fusion with lysosome. S1R seems to be involved in this step. (4) Formation of the autolysosome. (5) Degradation and recycling of the autolysosome contents.

**Figure 4 ijms-24-01971-f004:**
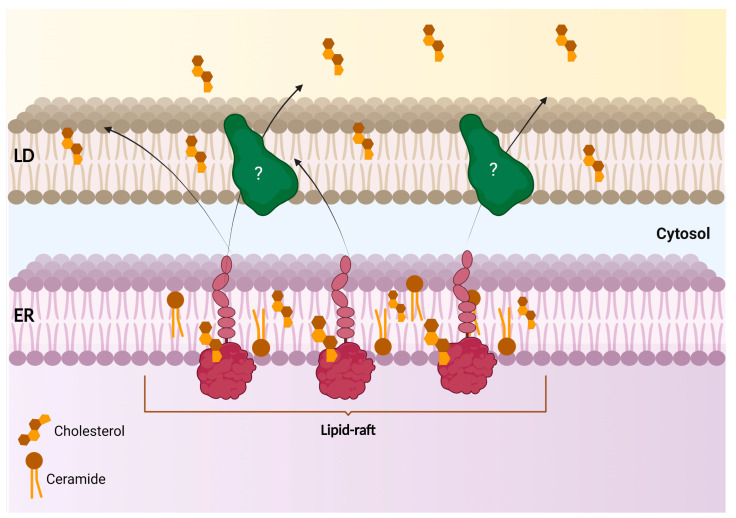
Scheme of S1R at the ER-LD contact site and its activity in lipid export from the ER and compartmentalization within LDs. S1R-enriched lipid raft at the ER controls lipid compartmentalization and export from the ER to lipid droplets.

**Figure 5 ijms-24-01971-f005:**
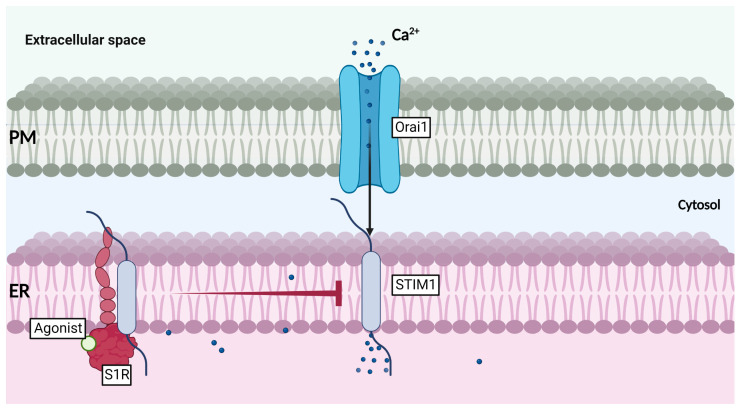
Function of S1R at the ER-PM junctions in ER-calcium entry.

**Figure 6 ijms-24-01971-f006:**
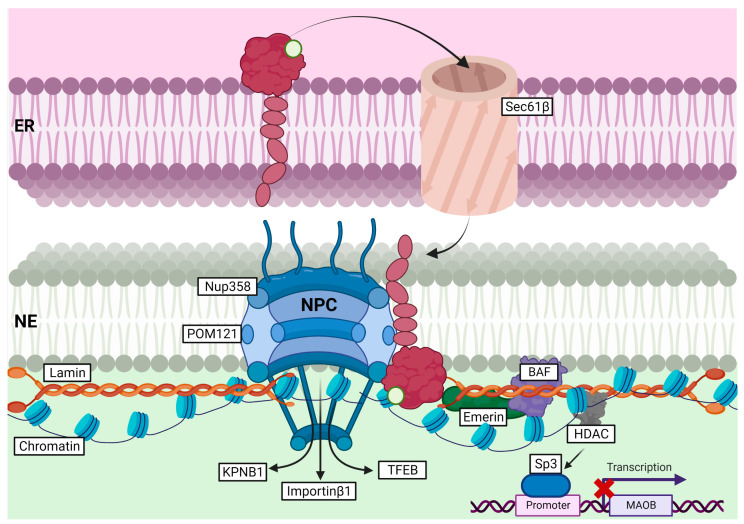
Role of S1R at the ER-NE contact site and its role in nucleocytoplasmic transport, stabilization of the NPC, and gene transcription.

## Data Availability

Not applicable.

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
