# Peer review of "Overview of Sigma-1R Subcellular Specific Biological Functions and Role in Neuroprotection"

_ijms, 2023, doi:10.3390/ijms24031971_

Round 1
Reviewer 1 Report
In this paper the authors make a review of the participation of Sigma1 receptor (S1R) in some cellular pathways contributing to neurodegenerative processes.
It is worth to note that several reviews on the same topic have been published in the last years, but the S1R is still receiving considerable attention in the biomedical field.
Comments
- The Abstract should be rewritten to give a clearer summary of the contents of the paper. Sigma1 receptor abbreviation S1R is used but not described.
- Section 2 is quite comprehensive of the implication of S1R in different cellular locations and homeostatic pathways. The schemas are simple but help to illustrate the text.
- Section 3, on the contrary, is too short and does not cover but a small part of the knowledge on S1R in motoneuron diseases. It mainly focuses on pridopidine. I suggest to eliminate this Section 3, so the review will improve in coherence and balance.
- Conclusions should be Section 4 (3 if the previous comment is attended).
Author Response
In this paper the authors make a review of the participation of Sigma1 receptor (S1R) in some cellular pathways contributing to neurodegenerative processes.
It is worth to note that several reviews on the same topic have been published in the last years, but the S1R is still receiving considerable attention in the biomedical field.
We agree that many reviews have been written on this topic. In this review, we decided to exploit this subject using a different approach. In fact, instead of writing a list of facts regarding S1R activity and role in neurodegeneration, we linked S1R functions to a specific subcellular compartment and mechanisms and connected it to a pathology. We think that using this original approach could benefit the field and refresh our way of thinking about S1R as a biological target and our ways to develop new therapeutics (e.g. Delivery to a subcellular specific compound).
Comments
- The Abstract should be rewritten to give a clearer summary of the contents of the paper. Sigma1 receptor abbreviation S1R is used but not described.
As recommended by the reviewer, we modified the abstract and described S1R abbreviation.
- Section 2 is quite comprehensive of the implication of S1R in different cellular locations and homeostatic pathways. The schemas are simple but help to illustrate the text.
- Section 3, on the contrary, is too short and does not cover but a small part of the knowledge on S1R in motoneuron diseases. It mainly focuses on pridopidine. I suggest to eliminate this Section 3, so the review will improve in coherence and balance.
As suggested, we removed the third section, but kept some of the therapeutics information that we combined with the conclusion in the third section.
- Conclusions should be Section 4 (3 if the previous comment is attended).
We changed the section identification.
Reviewer 2 Report
Present manuscript provides comprehensive review about Sigma-1R functions with a focus on neurodegenerative diseases. Despite the fact that it seems to be interesting there are some items that have to be either included or modified.
Authors do not describe expression of Sigma-1R in CNS. Is it uniform or is the expression of Sigma-1R localised in specific cell populations?
I do not agree with authors that: Failure of ERAD degradation is known to induce UPR activation. This was not shown so far despite the expected crosstalk between UPR and ERAD (Hwang and Qi, Trends Biochem Sci. 2018 Aug;43(8):593-605. doi: 10.1016/j.tibs.2018.06.005) and in general, ERAD is considered as a downstream response to ER stress/UPR. Although deletion of either SEL1L or HRD1 in various tissues and cell types resulted in a profound 10–100-fold increase in IRE1α protein, the level of phosphorylated IRE1α as well as its RNase enzymatic activity as measured by XBP1 mRNA splicing were moderately elevated in ERAD-deficient cells under basal conditions (Sun S et al. Nat. Cell Biol 17, 1546–1555). In turn, some essential components of ERAD (e.g. HRD1 or Derlin-3) are controled via IRE1 (Yamamoto et al. J Biochem. 2008 Oct;144(4):477-86. doi: 10.1093/jb/mvn091; Dibdiakova et al. Neurol Res. 2019 Feb;41(2):177-188. doi: 10.1080/01616412.2018.1547856.) or ATF6 (Belmont et al. Circ Res. 2010 Feb 5;106(2):307-16. doi: 10.1161/CIRCRESAHA.109.203901.) arm of UPR.
Author Response
Present manuscript provides comprehensive review about Sigma-1R functions with a focus on neurodegenerative diseases. Despite the fact that it seems to be interesting there are some items that have to be either included or modified.
Authors do not describe expression of Sigma-1R in CNS. Is it uniform or is the expression of Sigma-1R localised in specific cell populations?
We thank the reviewer for this comment. As mentioned in the introduction, we added cell-specific information regarding S1R expression in brain-specific cells.
I do not agree with authors that: Failure of ERAD degradation is known to induce UPR activation. This was not shown so far despite the expected crosstalk between UPR and ERAD (Hwang and Qi, Trends Biochem Sci. 2018 Aug;43(8):593-605. doi: 10.1016/j.tibs.2018.06.005) and in general, ERAD is considered as a downstream response to ER stress/UPR. Although deletion of either SEL1L or HRD1 in various tissues and cell types resulted in a profound 10–100-fold increase in IRE1α protein, the level of phosphorylated IRE1α as well as its RNase enzymatic activity as measured by XBP1 mRNA splicing were moderately elevated in ERAD-deficient cells under basal conditions (Sun S et al. Nat. Cell Biol 17, 1546–1555). In turn, some essential components of ERAD (e.g. HRD1 or Derlin-3) are controled via IRE1 (Yamamoto et al. J Biochem. 2008 Oct;144(4):477-86. doi: 10.1093/jb/mvn091; Dibdiakova et al. Neurol Res. 2019 Feb;41(2):177-188. doi: 10.1080/01616412.2018.1547856.) or ATF6 (Belmont et al. Circ Res. 2010 Feb 5;106(2):307-16. doi: 10.1161/CIRCRESAHA.109.203901.) arm of UPR.
As recommended, we removed this statement and thanked the reviewer for his thorough explanation.